# Implementation of ANN-Based Auto-Adjustable for a Pneumatic Servo System Embedded on FPGA

**DOI:** 10.3390/mi13060890

**Published:** 2022-05-31

**Authors:** Marco-Antonio Cabrera-Rufino, Juan-Manuel Ramos-Arreguín, Juvenal Rodríguez-Reséndiz, Efren Gorrostieta-Hurtado, Marco-Antonio Aceves-Fernandez

**Affiliations:** Facultad de Ingeniería, Universidad Autónoma de Querétaro, Cerro de las Campanas, Las Campanas, Queretaro 76010, Mexico; juvenal@uaq.edu.mx (J.R.-R.); efrengorrostieta@gmail.com (E.G.-H.); marco.aceves@gmail.com (M.-A.A.-F.)

**Keywords:** robot arm, pneumatic actuators, neural network, FPGA, embedded, neuro-PID, control

## Abstract

Artificial intelligence techniques for pneumatic robot manipulators have become of deep interest in industrial applications, such as non-high voltage environments, clean operations, and high power-to-weight ratio tasks. The principal advantages of this type of actuator are the implementation of clean energies, low cost, and easy maintenance. The disadvantages of working with pneumatic actuators are that they have non-linear characteristics. This paper proposes an intelligent controller embedded in a programmable logic device to minimize the non-linearities of the air behavior into a 3-degrees-of-freedom robot with pneumatic actuators. In this case, the device is suitable due to several electric valves, direct current motors signals, automatic controllers, and several neural networks. For every degree of freedom, three neurons adjust the gains for each controller. The learning process is constantly tuning the gain value to reach the minimum of the mean square error. Results plot a more appropriate behavior for a transitive time when the neurons work with the automatic controllers with a minimum mean error of ±1.2 mm.

## 1. Introduction

The employment of pneumatic actuators in the robotic field has great importance in the manufacturing industry and the modern design of controllers [1,2]. Its significant advantages are a low-cost, lightweight, and simple design. The disadvantages are the highly non-linear behavior of the pneumatic actuators due to having the air as a means of generating force, e.g., the flow of air into the valves is not uniform, friction in the joints, and delay in air propagation, among others [3,4,5,6,7].

Artificial neural networks (ANN), fuzzy logic, genetic algorithms, and others are practical when the mathematical model of the dynamic system is highly complex, highly non-linear, or impossible to know due to the lack of knowledge, except for a few known variables [8,9,10,11,12,13].

Programmable hardware devices, like FPGA, are suitable for hardware implementation of neural networks and PID controllers. They have the advantage of better accuracy, repeatability, and lower noise sensitivity. Also, it is compatible with other types of pre-processors [14,15,16,17].

This work aims to position a 3 DoF robot arm with pneumatic actuators and implement an ANN-based auto-adjustable gain tuner for PID controllers. The main contribution is an algorithm proposal to control each DoF of the pneumatic actuators embedded in reconfigurable hardware.

The pneumatic robot arm has six DoF, as presented in Figure 1, where $\theta_{1}$, $\theta_{2}$, and $\theta_{3}$ are the respective angle for each DoF. The moving forces are supplied by one pneumatic motor and two pneumatic pistons.

To convert a PID design into its VHDL code, we implement Octave software for this work. For example, we set PID gains as integer inputs, and the output is in binary fixed-point format.

### Findings

Table 1 shows a brief comparison of the implementations and contributions of state of the art. It mentions the hardware-implemented, and all these works are focused on pneumatic actuators.

The prototype architecture proposed has several problems, such as frictions in the joints and the non-linearity inevitably produced by the compressibility of the air. Airflow is not constant and experiences a delay in propagating through the system [3]. The lack of precision could cause catastrophic events that would affect the robot’s task and even harm workers [20].

The proposed robot controller remarkably has a slightly more accurate performance, adequately reducing alterations because of the non-linearities of the pneumatic actuators.

The order of this work is as follows: Section 2 deeply describes the materials and methodology, including the hardware description of the PID controller, the configuration of the encoder, and mainly the ANN. Section 3 presents the results obtained for the PID controllers of the motors and the ANN auto-adjustable gains for the pneumatic actuators controller. Section 4 is the discussion of results and their limitations. Finally, Section 5 gives the conclusions of this investigation.

## 2. Materials and Methods

This section presents the materials and methodology used in this work, such as inverse kinematics, the instrumentation of the encoders, the design of PID controllers and artificial neural networks, and some mathematical tools that helped us develop the before mentioned.

### 2.1. Materials

In this work, we used 24-VDC and 12-VDC power supplies. An FPGA was the core of this project and also a laptop for data acquisition, Octave script development, and FPGA programming. Eighteen optocouplers isolated the FPGA control signals from the motor drivers to prevent electric noise, and six solid-state relays activated the 12-VDC electro-valves. The motors adjusted the airflow through the pneumatic actuators. The pneumatic electro-valves controlled the direction movement of the links of the robot. Figure 2 illustrates all these components

### 2.2. Inverse Kinematics of a 3 DoF Robot

Figure 3 depicts a geometrical model of a 3-DoF robot, and it provides information to compute the final position $\left(x_{1},y_{1},z_{1}\right)$ of the robot [18], where:$l_{1}$, $l_{2}$, $l_{3}$→ are the links of the robot.$\theta_{1}$→ is the angle of link $l_{1}$ with respect to *Y* axis.$\theta_{2}$→ is the angle of link $l_{2}$ with respect to XY axis.$\theta_{3}$→ is the angle of link $l_{3}$ with respect to XY axis.

The inverse kinematic calculated the value of the angles $\theta_{1}$, $\theta_{2}$, and $\theta_{3}$ needed to get the final position required $\left(x_{1},y_{1},z_{1}\right)$. The Equations (Equation 1)–(Equation 3) represent the inverse kinematics of the robot [18].
(1)$$\theta_{1}=tan^{-1}\left(\frac{y_{1}}{x_{1}}\right)$$
(2)$$\theta_{2}=cos^{-1}\left(\frac{x^{2}+{\left(z-l_{1}\right)}^{2}+l_{2}^{2}-l_{3}^{2}}{2l_{2}\sqrt{x^{2}+{\left(z-l_{1}\right)}^{2}}}\right)+sin^{-1}\frac{z-l_{1}}{\sqrt{x^{2}+{\left(z-l_{1}\right)}^{2}}}$$
(3)$$\begin{array}{c} \theta_{3}=\theta_{2}+cos^{-1}\left(\frac{l_{3}^{2}+l_{2}^{2}-x^{2}-{\left(z-l_{1}\right)}^{2}}{2l_{3}l_{2}}\right)-180^{\circ } \end{array}$$

### 2.3. The Pneumatic System

Each DoF consists of one 5/3 electro-pneumatic valve, two airflow valves, two DC motors, and one incremental encoder that allowed us to get the arm position. In Figure 4, the schematic diagram of each DoF is drawn.

### 2.4. Structure Prototype for Airflow Control

We designed a mechanical base structure to couple the DC Motors to the airflow valves. A bell-type piece was attached to the motor stem. Moreover, it contains an internal hexagonal hole to fit the valve connection. Figure 5 depicts the integration of the mentioned parts.

### 2.5. Implemented Algorithm for Binarization of Variables

We developed Octave scripts to write VHDL code. They binarized numeric variables of logic signals, variables and constants, and LUTs faster. More helpful information about this is in [21].

For the binarization process, we used the fixed-point method. A fixed-point number in base-2 format is in (Equation 4).
(4)$${\left(\cdots i_{2}i_{1}i_{0}\cdot f_{-1}f_{-2}f_{-3}\cdots \right)}_{2}$$
which is converted to a decimal number as in Equation (Equation 5).
(5)$$\cdots i_{2}\cdot 2^{2}+i_{1}\cdot 2^{1}+i_{0}\cdot 2^{0}+f_{-1}\cdot 2^{-1}+f_{-2}\cdot 2^{-2}+f_{-3}\cdot 2^{-3}\cdots$$

We wrote the Algorithm 1 on a laptop. The input is an integer or fractional variable. The same variable returned as a fixed-point output. bf=i+f, where *i* is the number of bits for the integer part, *f* for the fractional part, and bf is the total of bits used for that variable.
**Algorithm 1:** Binarization of a fixed-point variable. $Aux=variable\times 2^{f}$  **if**
Aux<0
**then**
  $Aux=floor(2^{bf}+Aux)$ **else**  Aux=floor(Aux) **end if**  **for**
i=bf:−1:1
**do**  $Coeff\_binary\left(i\right)=Aux\%2$  Aux=floor(Auxiliar/2) **end for** $Aux=variable\times 2^{f}$

The *Coeff_binary* variable is a matrix with 1×bf size, *bf* is the number of zeros and ones stored in the matrix.

### 2.6. Design of the PID Algorithm

The continuous-time PID controller is given in (Equation 6). Its development and analysis are described in [22].
(6)$$u\left(t\right)=K_{p}\left[e\left(t\right)+\frac{1}{T_{i}}\int_{0}^{t}e\left(\tau \right)d\tau +T_{d}\frac{de(t)}{dt}\right]$$
where u(t) is the control variable, and e(t) is the error. The error is the difference between the reference w(t) and the output y(t). In Figure 6 a control system with the previous specifications is illustrated. The fundamental parameters of the PID controller are $K_{p}$, $T_{i}$, and $T_{d}$.

The Laplace transform converts the Equations (Equation 6) to (Equation 7).
(7)$$U\left(s\right)=K_{p}\left[1+\frac{1}{T_{i}s}+T_{d}s\right]E\left(s\right)$$
where *s* typically represents the Laplace transform operator. From (Equation 7), the transfer function G(s) is in (Equation 8).
(8)$$G\left(s\right)=\frac{U(s)}{E(s)}=K_{p}\left[1+\frac{1}{T_{i}s}+T_{d}s\right]$$

The proportional, derivative, and integral components in (Equation 6) are discretized to get the PID controller. $T_{0}$ is the sample period. For this work, we use $T_{0}=10$ ms, and Equation (Equation 9) represents the derivative error.
(9)$$\frac{de}{dt}\approx \frac{e(k)-e(k-1)}{T_{0}}=\frac{\Delta e(k)}{T_{0}}$$
where e(k) is the error at the *k*-th sampling time, i.e., at $t=kT_{0}$. The most convenient way to do the integral is by summing. Hence we approximate the continuous-time function by sampling using the direct trapezoidal method (see Figure 7). The integration of the error is computed with (Equation 10), and the discrete-time PID controller is (Equation 11).
(10)$$\int_{0}^{t}e\left(\tau \right)d\tau \approx T_{0}\sum_{i=1}^{k}e\left(i-1\right)$$
(11)$$u\left(k\right)=K_{p}\left\{e\left(k\right)+\frac{T_{0}}{T_{i}}\sum_{i=1}^{k}e\left(i-1\right)+\frac{T_{d}}{T_{0}}\left[e(k)-e(k-1)\right]\right\}$$

The Equation (Equation 11) reduce the computational time processing on the FPGA, resulting in (Equation 12). The constants $q_{1}$, $q_{2}$, and $q_{3}$ in Equation (Equation 13) are computed before the PID time process.
(12)$$u\left(k\right)=q_{1}e\left(k\right)+q_{2}e\left(k-1\right)+q_{3}e\left(k-2\right)+u\left(k-1\right)$$
(13)$$\begin{array}{cc} q_{1} & =K_{p}\left(1+\frac{T_{0}}{2T_{i}}+\frac{T_{d}}{T_{0}}\right)\\ q_{2} & =-K_{p}\left(1-\frac{T_{0}}{2T_{i}}+\frac{2T_{d}}{T_{i}}\right)\\ q_{3} & =K_{p}\frac{T_{d}}{T_{0}} \end{array}$$

Based on (Equation 12), we performed an Octave script to generate the VHDL code for a PID controller. Figure 8 shows the architecture resulting, where the input signals are reset (*RST*), clock (*CLK*), setpoint (*Sp*), sampling time (*Ts*), actual position (*Xin*), and the constant values ($q_{1}$, $q_{2}$, $q_{3}$). The output signal is *yk* and the sampling time was 10 ms.

Figure 8 is the PID block diagram for hardware configuration; the Register_PP_n module loads the input signal every period to compute the output *yk*. A subtraction module was applied to compute the actual error ek=sp−xin; the $q_{1}$, $q_{2}$ and $q_{3}$ signals are the controller constants. An adder module is used to compute $yk_{r}aw=q_{1}e_{k}+q_{2}e_{k2}+q_{3}e_{k3}+y_{k1}$, and four multipliers modules are implemented. The controller output signal yk is set to 18 bits.

### 2.7. PID Controller for the Airflow Valve

The arm position and speed are essential to perform the proper airflow control. The DC motors are linear systems that are practical to open or close the valve at the desired opening level to get the airflow required.

Based on Equations (Equation 12) and (Equation 13), Figure 9 shows the arm position and speed control proposal, where $W_{p}$ is the desired position, $e_{p}$ is the error position, $u_{p}$ is the control signal. We used a saturation block where $W_{v}$ is the saturation value and the maximum velocity desired. $e_{v}$ is the speed error, and $u_{v}$ is the control signal for the DC motor. The motor encoder sends position and speed feedback to the controller.

### 2.8. Encoder Instrumentation

The quadrature encoder module has two inputs, Channel A and B. DIR output signal gives the direction of the motor. ENA is active when a state changes on channels A and B. We implemented six flip-flops type D to catch the signals, one XOR Gate with two inputs for DIR output and another XOR gate with four inputs for ENA output. Figure 10 displays the architecture and the corresponding truth table of the encoder module.

Figure 11 presents the quadrature encoder instrumentation, written in Octave software. The motor channel inputs are A and B. Ts is the period signal. We set a counter for the pulses from the encoders for the speed. When the $T_{s}$ signal activates the rising edge module, the counter value is saved, and it is reset to start over. In Figure 11, outputs are drawn. The vel_nom and pos_nom are the raw values of the velocity and position. These have 16 bits.

The third output signal represents the normalized velocity. We obtained the value of the constant res_vel_norm in (Equation 14). Where $v_{MAX}=108RPM$ and $N_{v}=3432$ ppr, converted to 18 bits size. The vel_norm is the output of the normalized velocity given in the Equation (Equation 15).
(14)$$res\_vel\_norm=\frac{60}{V_{MAX}\times N_{v}\times T_{s}}$$
(15)vel_norm=res_vel_norm×vel_nom

For the normalization of the position in 18 bits, we considered the maximum of 10 rotations of the flow control valve for its constant given in Equation (Equation 16). The pos_norm is the output of the normalized position in Equation (Equation 17).
(16)$$res\_norm=\frac{1}{P_{MAX}\times N_{v}}\times pos\_nom$$
(17)pos_norm=res_pos_nom×res_pos

### 2.9. Neural Network Design

Artificial neurons are suitable for non-linear systems, providing continuous outputs, gathering signals available on their inputs, and assembling them according to their operational and intuitive activation. Figure 12 illustrates each neuron of a network. The multiple input signals coming from the external environment (specific application) are represented by the set $\left\{x_{1},x_{2},x_{3},\cdots ,x_{n}\right\}$ [23].

The weighting carried out by the synaptic junctions of the network is implemented on the artificial neuron as a set of weights $\left\{w_{1},w_{2},\cdots ,w_{n}\right\}$. Analogously, the relevance of each of the $\left\{x_{i}\right\}$ inputs is measured by multiplying them by their corresponding weight $\left\{w_{i}\right\}$, then weighting all the external information arriving at the neuron. Therefore, the neurons output is denoted by *y*, representing the weighted sum of its inputs. Equations (Equation 18) and (Equation 19) synthesize the result produced by the artificial neuron [24].
(18)$$u=\sum_{i=1}^{n}w_{i}x_{i}-\theta$$
(19)$$y=g\left(u\right)$$

We trained the neurons by the method known as the backpropagation algorithm. By this method, we got the mean square error of the Equation (Equation 20).
(20)$$E\left(t\right)=\frac{1}{2}\sum e^{2}\left(t\right)$$

### 2.10. Neuron Learning Algorithm

The specific purpose of this design is to properly use perceptron neurons to tune the variables $k_{p}$, $T_{i}$, and $T_{d}$ of a digital PID controller in Equation (Equation 13). The objective is the mean square error, with a ±0.5 tuning on each variable. These neurons are constantly learning. And the steps for the algorithm are the following [23].
The error signals *e* and Δe in (Equation 21) and (Equation 22) are computed from the desired position for each of the robot joints $q=(\theta_{1},\theta_{2},\theta_{3})$, and the actual position being measured of the system $r=(x_{1},x_{2},x_{3})$.
(21)e(t)=y(t)−w(t)
(22)Δe(t)=e(t−1)+e(t)The error and derivative error are the inputs for each neuron. The description of this is in the Equation (Equation 23).
(23)$$X\left(t\right)=\left[\begin{array}{cc} e(t) & \Delta e(t) \end{array}\right]$$Initialize *w* with small random values. For example w=0.1. We defined them in (Equation 24), and it represents the sum of the neuron weights in the first layer.We set the learning rate to η=0.99. And the following steps are repeated permanently, and the neuron is constantly learning.The *s* variable is computed in (Equation 24).
(24)$$s=w_{1}e\left(t\right)+w_{2}\Delta e\left(t\right)$$The activation function, a sigmoid, of the intermediate neurons in (Equation 25) is computed, and Figure 13 is its representation.
(25)$$h=\frac{0.5}{1+e^{-s}}$$The adaptive Equations (Equation 26) and (Equation 27) allowed the proportional coefficient $K_{p}$ values to be adjusted. We used a similar development to find the adjustment equations for $T_{d}$ and $T_{i}$. The value η is the learning coefficient of the neural network and was 0.9.
(26)$$v\left(t+1\right)=v\left(t\right)+\eta e{\left(t\right)}^{2}h$$
(27)$$w_{j}\left(t+1\right)=w_{j}\left(t\right)+\eta e{\left(t\right)}^{2}vh\left(1-h\right)x_{j}$$The proportional gain is denoted by the Equations (Equation 28) and (Equation 29), where *v* is the weight of the last neuron and *h* is the activation function.
(28)$$\Delta K_{p}=vh$$
(29)$$K_{p}=K_{p}+\Delta K_{p}$$Return to step 4.

Figure 14 shows the design of the neurons for $k_{p}$, and identical block diagrams are implemented for $T_{d}$ and $T_{i}$. And they need to be initialized in this structural design. $K_{pc}$, $T_{ic}$, and $T_{dc}$ are the constant inputs. The Rising_edge and Latch modules detect the first pulse of the period signal $T_{0}$, adding these variables to a summand that stores the initial value and updates the value of the kp, ti, and td variables. For $\Delta e_{k}$, a subtraction with a delay module is implemented to achieve Δe(k)=e(k−1)−e(k), for *s* signal two multipliers with an adder module were implemented for (Equation 24). To achieve (Equation 25) a LUT was implemented, and it contains $2^{10}$ values between −1 to 1.

For *v* in Equation (Equation 25), a multiplication module with four inputs was implemented, and an adder with a period delay module to compute v(k+1). For the values of the weights $w_{1}$ and $w_{2}$ another four variables multiplier was needed, the $\eta e{\left(t\right)}^{2}h$ factor in (Equation 25) was repeated for (Equation 26). A single multiplier is needed to calculate (Equation 27). The value of $k_{p}$ is constantly tuning, and $k_{pc}$ is the initial value of $k_{p}$. To get $k_{p}$ an adder is implemented, summing $\Delta k_{}p$ every period.

Finally, $K_{p}$, $T_{i}$, and $T_{d}$ are computed to obtain $q_{0}$, $q_{1}$, and $q_{2}$ in Equation (Equation 13), Figure 15 is this conversion.

Figure 16 presents the general diagram of a DoF, as it is observed inverse kinematics compute the desire angle values for a particular position of the end effector. Though only the second DoF is drawn, identical architecture was designed for the first and last DoF. According to $e_{k}$ signed value, the direction of the valve was chosen.

The FPGA resources implemented in the project are observed in Table 2. We spent almost half of the resources, but the multipliers, that our project used 100%.

## 3. Results

This section, first shows the flow air control signals, position, and speed, including simulations and experimental results. Then, a graph with a PID and Neuro-PID experimental results for comparison. After that, we plotted the results of the 3 DoF, and we included error and derivative error signals graphs.

### 3.1. DC Motors Control for Airflow

As a first step, controlling the airflow valves is necessary. The VDC motors are the appropriate actuators due to their linear behavior and easy tuning. Figure 17 plots the simulations of the position and speed. When the system starts, the motor reaches the maximum allowed speed; it tends to be zero once it gets to the desired position. Figure 18 contains the experimental results, which are similar to those simulated. The sample timeairflows. As the air flow valves have ten spins maximum, the top position is 63 rads. The full speed of the motors is 210 RPM, which means 22 rad/s.

We first considered a Texas Instruments DSP for the experimental results, which gave excellent results. However, PID controllers and neural networks needed a fast response due to the number of encoders. We decided to switch to the FPGA, a device that can perform all these processes in parallel. For these results, the position PID parameters are $q_{0}=0.1030$, $q_{1}=-0.0949$, and $q_{2}=0.003$, for speed $q_{0}=0.0770$, $q_{1}=-0.0140$, and $q_{2}=0.007$.

### 3.2. PID vs. Neuro-PID Controller

Once the robotic arm was configured, we performed tests with the designed controllers. For the PID and Neuro-PID gains, we are using the heuristic method, and the gain adjustment was performed by experimental tests. Figure 19 and the blue line is the classic PID controller. The red line is the Neuro-PID controller, which has a particular behavior with a faster response time, a little less overshoot, and reaches the steady-state before the typical PID. Table 3 displays the transient response specifications of these controllers.

We observed that the intelligent controller has some disturbances, possibly due to the adjustment of the variable $K_{p}$. For this test, we set $K_{p}=2.5$, $T_{i}=0.98$, and $T_{d}=0$. In addition, for this and the following tests, the position and velocity variables were normalized. The maximum value of the signals is equal to one; this helps to use the computational resources better.

### 3.3. Positioning for the 3 DoF Links

The graph of the first DoF is in Figure 20. This is the one controlled by the pneumatic motor. Within these tests, this was the easiest to tune. It performs well, even when moving in conjunction with the other two DoFs.

Figure 21 and Figure 22 plot the second and third DoFs. These are controlled by the pistons located on the robot. Their tuning was more complex than the first one. In the second DoF, there are disturbances, but it is possible to tune to the marked reference. The third one was the most difficult to implement.

Figure 23 plots the error signal in blue color, the derivative error in red color. These signals are important for the PID and Neural network modules, and the error at steady-state is 1.2 mm.

## 4. Discussion

The proposed model has been developed to be applied to a 3 DoF pneumatic robot. The model uses three optical encoders with 1000 pulses by turn. Therefore, the resolution of the movement arm is 6.28 mrad. A limitation of the model is that the arm only has 120° to turn. That is due to the mechanical limitations.

The flow control is implemented with electrical motors coupled with flow control valves. Each motor for the air valve uses an encoder to know the state of the valve (open, close, and intermediate level). The DC motors use an electromagnetic encoder with 3432 pulses by turn. A limitation of the model is that we need two DC motors with encoders and their hardware to control them. In consequence, the model requires 6 DC motors with an encoder.

The implemented model must consider the hardware implementation for nine encoders, 3 PID control algorithms, 3 ANN algorithms, and the external logical signals used were 50 pins. We are using the following hardware of the FPGA D0-Nano board: 45% elements, 1671 registers, 32% pins, and 100% of the 9 bits multipliers.

In Section 3, according to Table 3 and Figure 19, we can observe that the Neuro-PID algorithm requires less rise time, overshoot, and settling time. The error is similar in both cases, but we consider that the ANN-PID algorithm has a better behavior than a simple PID. The limitation of the algorithm is the disturbances observed in Figure 19 on the Neuro-PID graph.

The disturbances in ANN-PID are due to the self-tuning process of the ANN algorithm to obtain the appropriate gains for the corresponding arm. One of the advantages of this process is that in the event of any disturbance that occurs in the arm positioning, the system will automatically respond to adjust itself. Moreover, the system has the advantage that if you change the weight being moved or add weight to one of the arms, the system responds by adjusting gains accordingly.

## 5. Conclusions

This work had three phases. The first one was the airflow control through the pneumatic system; although we did not control the air pressure directly, the positioning of the motors is adequate for its regulation. We performed the first tests on development boards based on microcontrollers. The motor speed control is considered for this design to avoid abrupt airflows. Figure 18a, shows the position of DC motor control, with an error of less than 0.01 rad. Moreover, Figure 18b represents the speed control, with an error of ±0.5 rad/s. In this phase, the goal was achieved, With accurate position control and the speed controlled at slow or medium values.

The second phase was the control design for 1 DoF. This step was performed entirely in VHDL. Moreover, in this phase, the results in Figure 19 show a better performance when the neural network was active, improving the overshoot, response time, and reaching the steady state. We got a minimum error of 1.2 mm in the steady-state. The results in this phase were satisfactory.

The last phase is the implementation of the 3 DoFs. Results in Figure 20, Figure 21 and Figure 22 show that the last two DoF were positioned to the desired position and had an error of 2 mm. We concluded that these results were acceptable. In future work, We consider implementing algorithms with fuzzy logic, neuro-fuzzy, and ANFIS.

## Figures and Tables

**Figure 1 micromachines-13-00890-f001:**
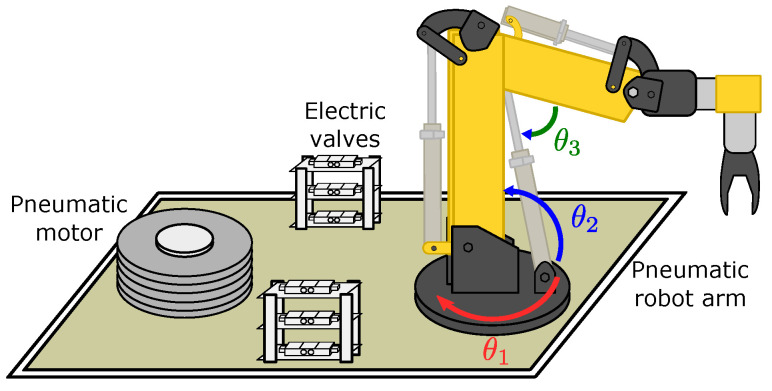
Pneumatic robot prototype with 6 DoF.

**Figure 2 micromachines-13-00890-f002:**
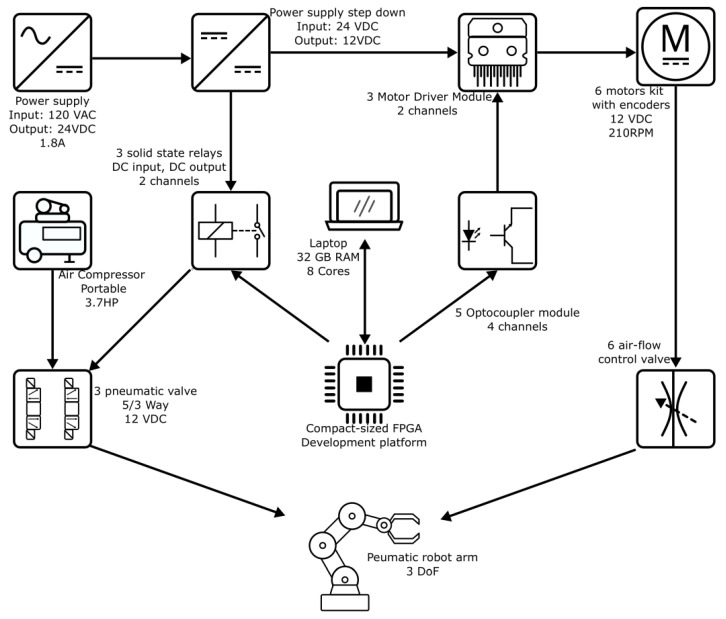
Diagram block of all materials used for this project.

**Figure 3 micromachines-13-00890-f003:**
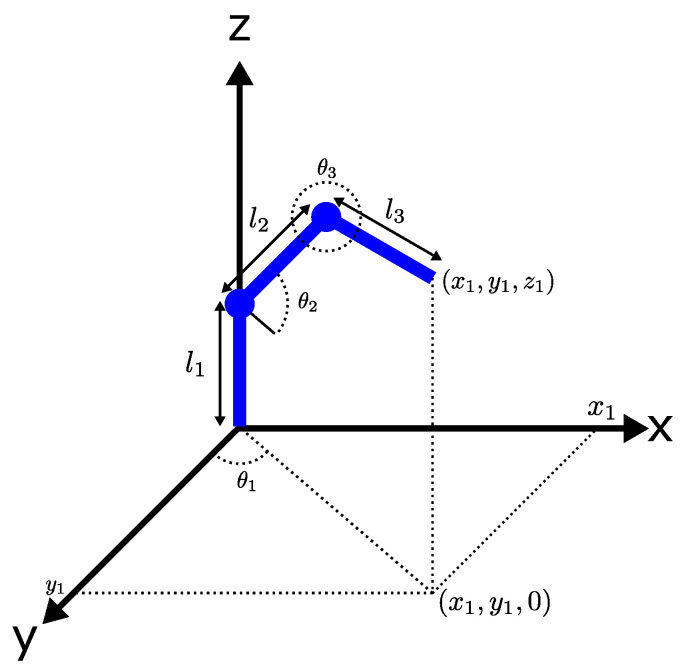
Geometrical model of a 3-DoF robot.

**Figure 4 micromachines-13-00890-f004:**
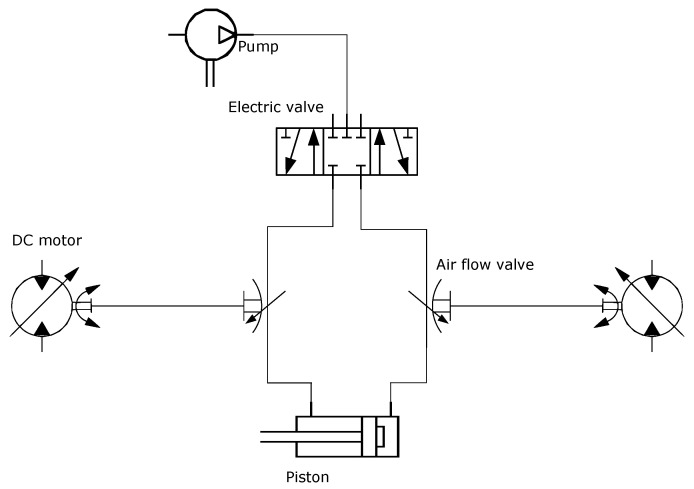
Schematic of one DoF pneumatic system.

**Figure 5 micromachines-13-00890-f005:**
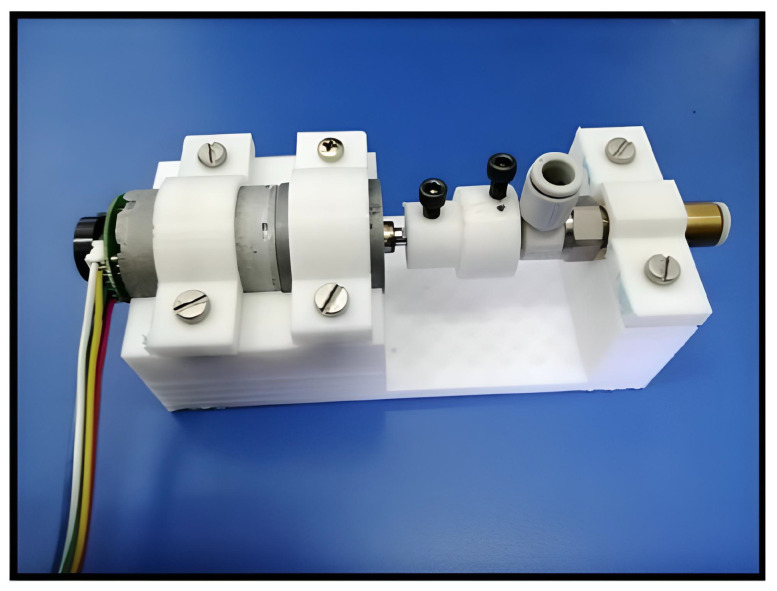
Airflow control system valve with plastic base for DC motors.

**Figure 6 micromachines-13-00890-f006:**
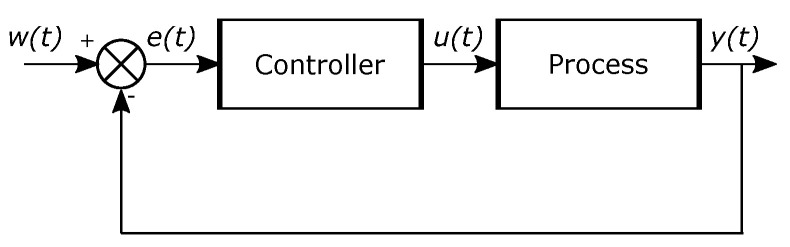
Block diagram of a control system [22].

**Figure 7 micromachines-13-00890-f007:**
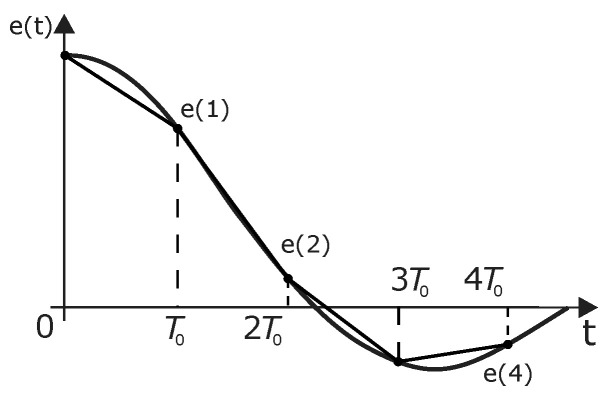
The forward trapezoidal method.

**Figure 8 micromachines-13-00890-f008:**
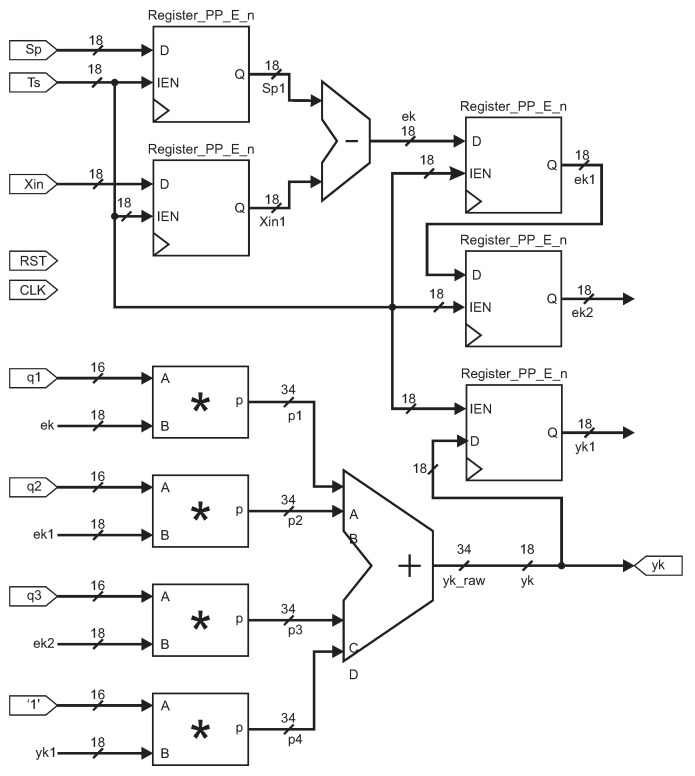
VHDL diagram block for PID controller. The symbol * in this figure represents a multiplier module.

**Figure 9 micromachines-13-00890-f009:**
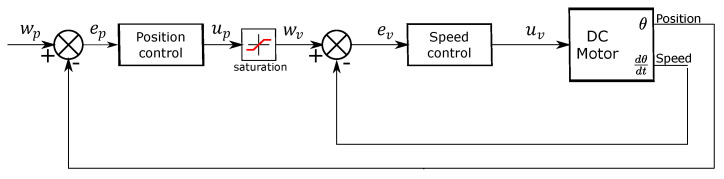
Diagram block for the motors PID controller.

**Figure 10 micromachines-13-00890-f010:**
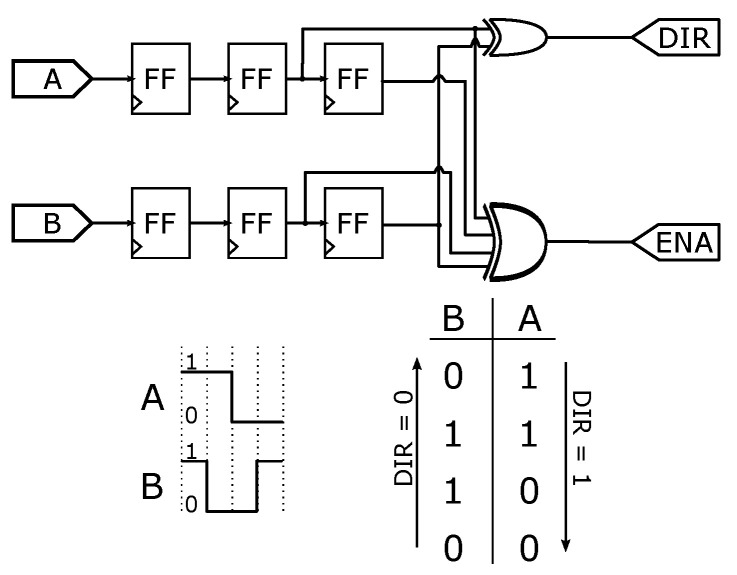
Encoder module functionality.

**Figure 11 micromachines-13-00890-f011:**
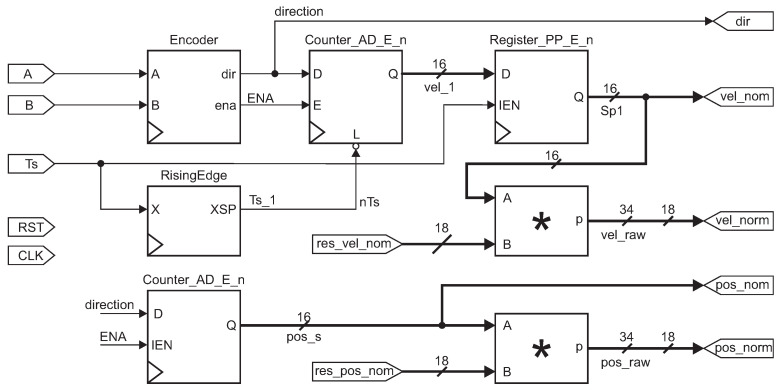
VHDL diagram block for the encoder configurations. The symbol * in this figure represents a multiplier module.

**Figure 12 micromachines-13-00890-f012:**
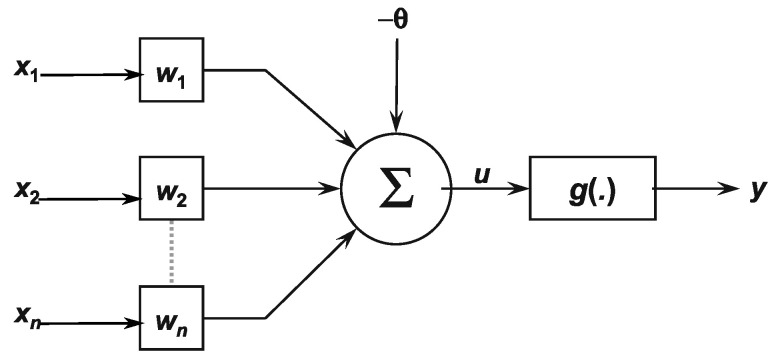
Artificial neural network.

**Figure 13 micromachines-13-00890-f013:**
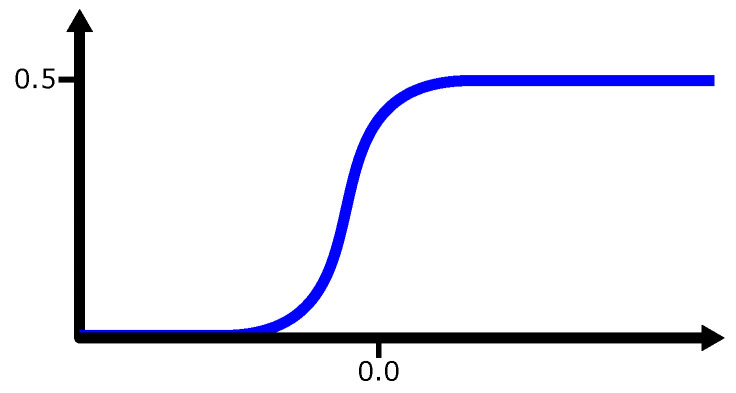
The sigmoid function.

**Figure 14 micromachines-13-00890-f014:**
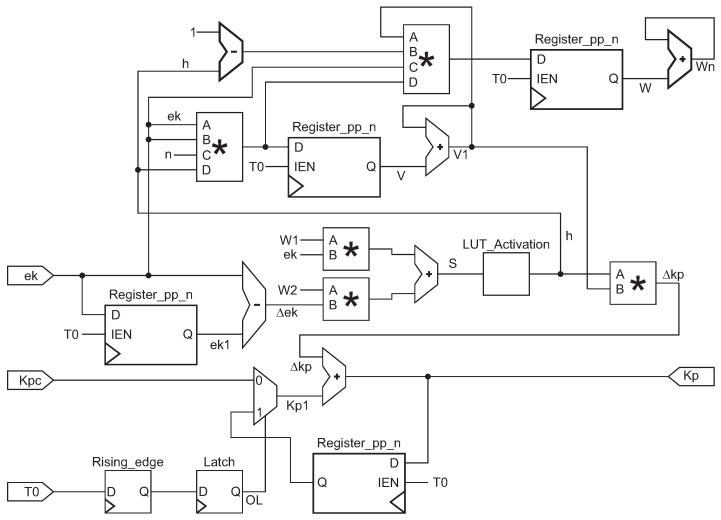
NNET module, neuro tuning variable for $k_{p}$. The symbol * in this figure represents a multiplier module.

**Figure 15 micromachines-13-00890-f015:**
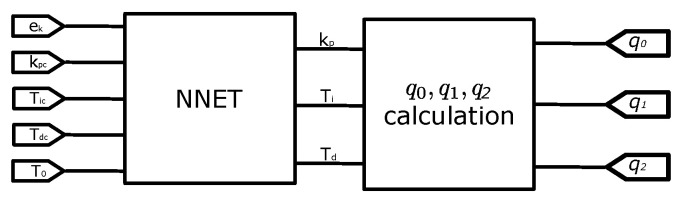
The calculation for $q_{0}$, $q_{1}$, and $q_{2}$.

**Figure 16 micromachines-13-00890-f016:**
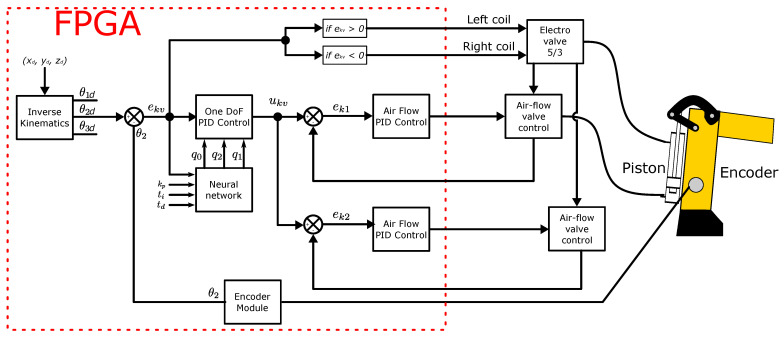
General diagram for a DoF.

**Figure 17 micromachines-13-00890-f017:**
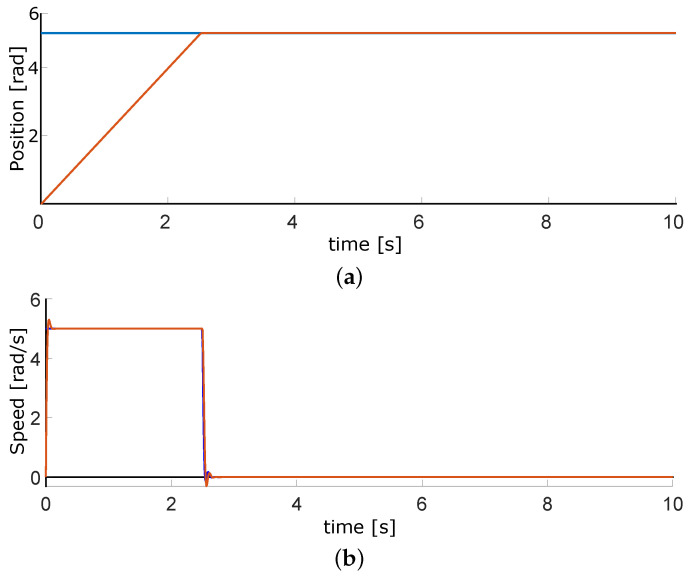
Simulated results for position and velocity control of DC motors where the blue lines are the setpoint and red ones the control signals. (**a**) Position plot. (**b**) Velocity plot.

**Figure 18 micromachines-13-00890-f018:**
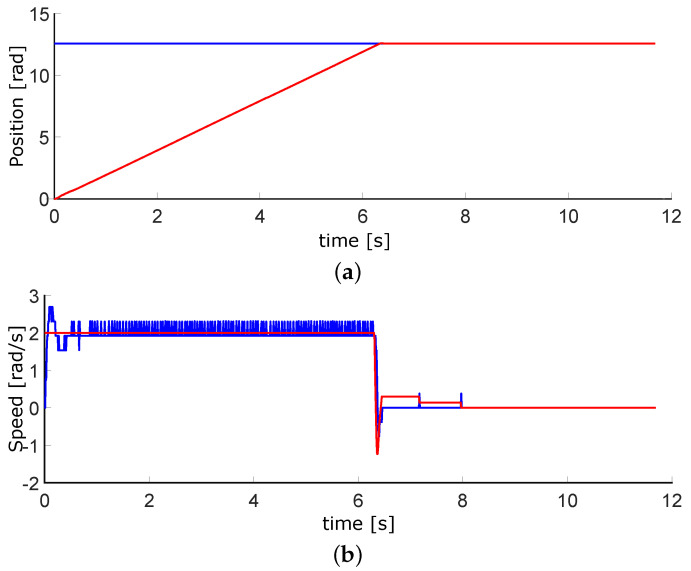
Experimental results for position and velocity control of DC motorswhere the blue lines are the setpoint and red ones the control signals. (**a**) Position plot. (**b**) Velocity plot.

**Figure 19 micromachines-13-00890-f019:**
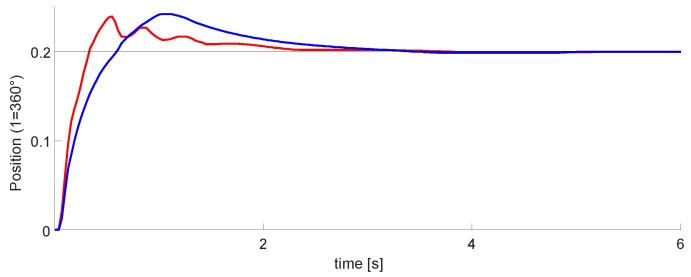
Experimental results for PID (blue) vs. Neuro-PID (red) controller for a DoF.

**Figure 20 micromachines-13-00890-f020:**
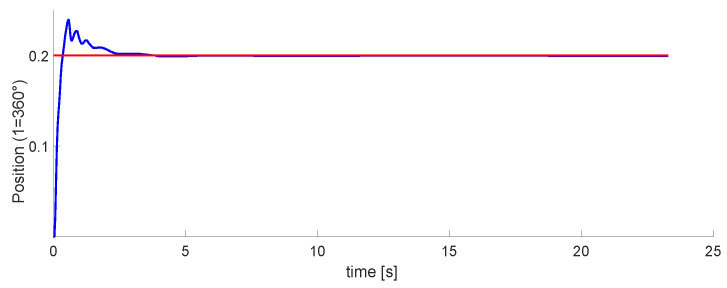
Experimental results for the first DoF. The Setpoint signal is colored in red, control signal in blue.

**Figure 21 micromachines-13-00890-f021:**
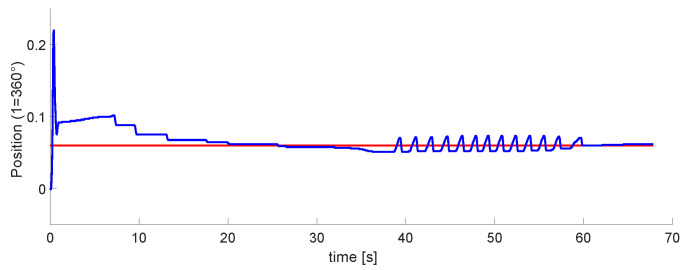
Experimental results for the second DoF. The Setpoint signal is colored in red, control signal in blue.

**Figure 22 micromachines-13-00890-f022:**
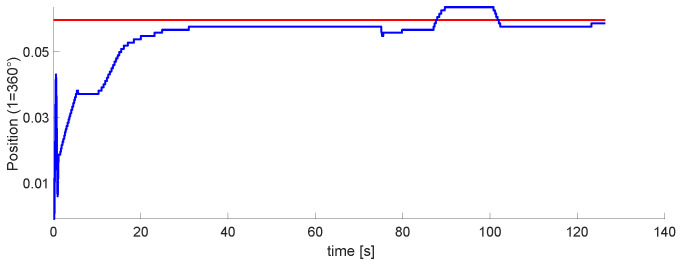
Experimental results for the third DoF. The Setpoint signal is colored in red, control signal in blue.

**Figure 23 micromachines-13-00890-f023:**
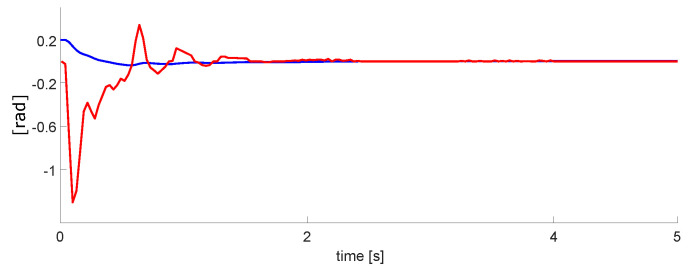
Experimental results for the error signal (blue) and derivative error (red) of a DoF.

**Table 1 micromachines-13-00890-t001:** State of the art.

Ref, Year	Implementation	Contribution
Sánchez, 2017 [18]	A micro-computer with a graphical simulator developed in C++ using OpenGL libraries	A simulated backpropagation artificial neural network controller for a two DoF pneumatic manipulator
Rousbeh, 2018 [6]	PCI-6602 DAQ board and a PC	The design of a controller and its implementation on a position-controlled rotary pneumatic actuator
Humaidi, 2020 [19]	Computer simulation in Matlab and Simulink	The design of a controller based on the Synergetic Control theory concept for a DoF robot arm powered by pneumatic actuators
Lin, 2021 [7]	myRio board and a PC with Labview	A PID controller and a high-order sliding-mode feedback controller for a 3 DoF pneumatic robot manipulator

**Table 2 micromachines-13-00890-t002:** Total resources of the hardware implementation and its quantities.

Total Resources	Quantity
Logic elements	10,134/22,320 (45%)
Registers	1671
Pins	50/154 (32%)
Multipliers 9 bits	132/132 (100%)

**Table 3 micromachines-13-00890-t003:** Transient response specifications of PID and Neuro-PID controllers.

	PID	Neuro-PID
	Kp=2.5,Td=0, and Ti=0.98	
Rise time [s]	0.61	0.33
Peak time [s]	1.07	0.52
Overshoot [mm]	39.1	36.3
Settling time [s]	4.38	2.37
steady-state-error [mm]	2.22	2.2

## Data Availability

Not applicable.

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
