# Peer review of "Implementation of ANN-Based Auto-Adjustable for a Pneumatic Servo System Embedded on FPGA"

_micromachines, 2022, doi:10.3390/mi13060890_

Round 1

Reviewer 1 Report

1. Typo in the Title: Autoadjuntable -> Autoadjustable. There are other typos throughout the paper, please cross-check.

2. Avoid using acronyms in abstract

3. State clearly research question and main contributions in the introduction section

4. Check the order of references, in introduction, line 25, [4,16-18]

5. Discuss the limitations of the proposed model.

6. Discuss the comparisons between PID and Neuro-PID in more details. Discuss the disturbances observed on Neuro-PID in Fig. 19.

7. Include more relevant recent references, as the bibliography could be more extensive. You can include the following three:

https://ieeexplore.ieee.org/abstract/document/9614876

https://link.springer.com/article/10.1007/s12652-021-02969-5

https://ieeexplore.ieee.org/abstract/document/9649985

Author Response

We appreciate the time that the reviewer takes to review this paper. We were following your suggestions to improve the work. The corrections according to your comments are presented.

  1. Typo in the Title: Autoadjuntable -> Autoadjustable. There are other typos throughout the paper, please cross-check.

We appreciate your time and comments about the typos. The paper has been cross-checked to eliminate it.

  1. Avoid using acronyms in abstract

We appreciate your time and comments about the acronyms in the abstract. The abstract has been corrected and eliminated all acronyms.

  1. State clearly research question and main contributions in the introduction section

We appreciate your time and comments about the Introduction section. We added the next paragraph in line 29. We are considering that the question is implicit in it.

“This work aims to position a 3 DoF robot arm with pneumatic actuators and implement an ANN-based auto-adjustable gain tuner for PID controllers. The main contribution is an algorithm proposal to control each DoF of the pneumatic actuators embedded on reconfigurable hardware.”

  1. Check the order of references, in introduction, line 25, [4,16-18]

We appreciate your time and comments about the references. All references were checked and corrected.

  1. Discuss the limitations of the proposed model.

We appreciate your time and comments about the proposed model. A paragraph discussion was added in line 252.

“The proposed model has been developed to be applied to a 3 DoF pneumatic robot. The model uses three optical encoders with 1000 pulses by turn. Therefore, the resolution of the movement arm is 6.28 mrad. A limitation of the model is that the arm only has 120° to turn. That is due to the mechanical limitations.

The flow control is implemented with electrical motors coupled with flow control valves. Each motor for the air valve uses an encoder to know the state of the valve (open, close, and intermediate level). The DC motors use an electromagnetic encoder with 3432 pulses by turn. A limitation of the model is that we need two DC motors with encoders and their hardware to control them. In consequence, the model requires 6 DC motors with an encoder.

The implemented model must consider the hardware implementation for nine encoders, 3 PID control algorithms, 3 ANN algorithms, and the external logical signals used were 50 pins. We are using the following hardware of the FPGA D0-Nano board: 45% elements, 1671 registers, 32% pins, and 100% of the 9 bits multipliers.”

  1. Discuss the comparisons between PID and Neuro-PID in more details. Discuss the disturbances observed on Neuro-PID in Fig. 19.

We appreciate your time and comments about the comparisons between PID and Neuro-PID. A discussion paragraph was added in line 267.

“In section results, according to table 3 and figure 19, we can observe that the Neuro-PID algorithm requires less rise time, overshoot, and settling time. The error is similar in both cases, but we consider that the ANN-PID algorithm has a better behavior than a simple PID. The limitation of the algorithm is the disturbances observed in figure 19 on the Neuro-PID graph.

The disturbances in ANN-PID are due to the self-tuning process of the ANN algorithm to obtain the appropriate gains for the corresponding arm. One of the advantages of this process is that in the event of any disturbance that occurs in the arm's position, the system will automatically respond to adjust itself. Also, the system has the advantage that if you change the weight being moved or add weight to one of the arms, the system responds by adjusting gains accordingly.”

  1. Include more relevant recent references, as the bibliography could be more extensive. You can include the following three:

We appreciate your time and comments about including more relevant references. We have included the next references:

https://ieeexplore.ieee.org/abstract/document/9614876 was included in line 24, as reference number [13].

https://link.springer.com/article/10.1007/s12652-021-02969-5 was included in line 28, as reference number [16]

https://ieeexplore.ieee.org/abstract/document/9649985 was included in line 16, as reference number [2]

Reviewer 2 Report

1. The title change is recommended as "Implementation of ANN-based Auto-adjustable Pneumatic Servo System using FPGA".

2. What criterion author followed to determine the Neuro PID gains in section 3.2? 

3. Authors must tabulate %OS, steady-state error, settling time and other related attributes in both PID and Neuro PID cases and compare them with each other for a different set of P, I, and D gains.

Author Response

We appreciate the time that the reviewer takes to review this paper. We were following your suggestions to improve the work. The corrections according to your comments are presented.

  1. The title change is recommended as "Implementation of ANN-based Auto-adjustable Pneumatic Servo System using FPGA".

We appreciate your time and comments about the title. We change the title to “Implementation of ANN-based Auto-adjustable for a Pneumatic Servo System embedded on FPGA”

  1. What criterion author followed to determine the Neuro PID gains in section 3.2?

We appreciate your time and comments about the criterion to determine the Neuro-PID gains. We are explaining it with the sentence between 230 and 231 lines.

“For the Neuro-PID gains, we are using the heuristic method and the gain adjustment was performed by experimental tests.”

  1. Authors must tabulate %OS, steady-state error, settling time and other related attributes in both PID and Neuro PID cases and compare them with each other for a different set of P, I, and D gains

We appreciate your time and comments about the table required. In this case, table 3 was added between lines 234 and 235.